# Determining cost and placement decisions for moderate complexity NAATs for tuberculosis drug susceptibility testing

**Akash Malhotra**[1], **Ryan Thompson**[1], **Margaretha De Vos**[2], **Anura David**[3], **Samuel Schumacher**[4], **Hojoon Sohn**[5]*

1 Department of Epidemiology, Johns Hopkins Bloomberg School of Public Health, Baltimore, MD, United States of America, 2 Foundation for Innovative New Diagnostics (FIND), Geneva, Switzerland, 3 University of the Witwatersrand, Johannesburg, South Africa, 4 World Health Organization (WHO), Geneva, Switzerland, 5 Department of Preventive Medicine, Seoul National University College of Medicine, Seoul, South Korea

* hsohn@snu.ac.kr

**Data Availability Statement:** All relevant data are within the paper and its Supporting Information files.

## Abstract

### Background

Access to drug resistant testing for tuberculosis (TB) remains a challenge in high burden countries. Recently, the World Health Organization approved the use of several moderate complexity automated nucleic acid amplification tests (MC-NAAT) that have performance profiles suitable for placement in a range of TB laboratory tiers to improve drug susceptibility tests (DST) coverage.

### Methods

We conducted cost analysis of two MC-NAATs with different testing throughput: Lower Throughput (LT, < 24 tests per run) and Higher Throughput (HT, upto 90+ tests per run) for placement in a hypothetical laboratory in a resource limited setting. We used per-test cost as the main indicator to assess 1) drivers of cost by resource types and 2) optimized levels of annual testing volumes for the respective MC-NAATs.

### Results

The base-case per test cost of $18.52 (range: $13.79 - $40.70) for LT test and $15.37 (range: $9.61 - $37.40) for HT test. Per test cost estimates were most sensitive to the number of testing days per week, followed by equipment costs and TB-specific workloads. In general, HT NAATs were cheaper at all testing volume levels, but at lower testing volumes (less than 2,000 per year) LT tests can be cheaper if the durability of the testing system is markedly better and/or procured equipment costs are lower than that of HT NAAT.

### Conclusion

Assuming equivalent performance and infrastructural needs, placement strategies for MC-NAATs need to be prioritized by laboratory system's operational factors, testing demands, and costs.

**Funding:** This work was funded from Seoul National University 800-20220288 through multi-donor core funding to Foundation for Innovative New Diagnostics (FIND). There was no specific funder from FIND as it was from multi-donor CORE funding.

**Competing interests:** The authors have declared that no competing interests exist.

## Introduction

In the past 10 years, around 3–4% of new tuberculosis (TB) cases and 18–21% of previously treated cases had multidrug-resistant- (MDR) or rifampicin-resistant- (RR) TB [1]. An estimated 157, 903 cases of MDR TB were detected globally in 2020 [1]. The rise of drug-resistant TB has led to increased demand for drug susceptibility testing (DST). DST coverage for TB increased globally from 50% in 2018 (1.7 million) to 71% in 2020 (2.1 million), with improvements in testing coverage in all 6 World Health Organization (WHO) regions [1]. Absence of key infrastructure, funding, and technological gaps in low- and middle-income countries (LMIC) leads to lower access to DST and treatment [2, 3].

Unprecedented scale-up of GeneXpert technology over the past decade in LMICs has offset this problem to some extent by improving TB detection, including for RR TB. While RR is an important surrogate marker for multi-drug resistant (MDR) TB [4], RR alone is not always adequate to initiate MDR-TB treatment, particularly in settings with high rifampicin (RIF) or isoniazid (INH) mono-resistance [5, 6]. Existing genotypic drug susceptibility testing (gDST) such as line probe assays (LPAs), e.g., GenoType MTBDR*plus* can test for RIF and INH resistance but lack performance and operational profiles suitable for lower-level health and laboratory systems where most clinical management occurs [7, 8].

Recently, several automated moderate complexity automated NAATs–Abbott RealTi*me* MTB and MTB RIF/INH (Abbott), FluoroType® MTBDR and FluoroType® MTB (Hain Lifescience/Bruker), BD MAX™ MDR-TB (Becton Dickinson) and cobas® MTB and cobas MTB-RIF/INH (Roche)–with capacity to test for INH and RIF resistance have been approved for use by the WHO [9]. These automated NAATs offer higher throughput capacity (single-run capacity: 24 to 96 samples) with comparable performance and lower infrastructural needs compared to WHO endorsed DSTs, thus being advantageous to both patients and programs [9, 10]. These NAATs are faster and less complex than phenotypic culture based drug susceptibility testing (DST) and line-probe assays (LPA), being largely automated after the sample preparation step [9]. Placement of these NAATs strategically in relation to the point-of-care can help improve DST coverage and DR-TB patient management [11]. However, successful implementation of these NAATs is dependent on a robust sample transport network and a high population density [9]. This may potentially lead to exclusion of low population communities with poor infrastructure.

Understanding the costs and associated cost drivers of these new technologies is critical in developing placement strategies. In this study, we assessed costs of two moderate complexity automated NAATs with different throughput capacity as routine DST in a hypothetical setting in a LMIC. Our main objectives were to determine the conditions under which per-test costs can be optimized for each NAAT and to evaluate the drivers of cost.

## Methods

### Intervention

Time-and-motion (TAM) study was carried out as part of an external laboratory evaluation by FIND, the global alliance for diagnostics, and the Department of Molecular Medicine and Haemotology (University of Witwatersrand) in South Africa [11]. We defined Low Throughput (LT) system as those test platforms with a capacity to test up to 24 samples per batch (i.e., singe test run). High Throughput (HT) test was defined as a system with a capacity of test up to 96 samples per batch, and which required reflex testing for confirmatory diagnosis. Based on the review of the full laboratory workflow of LT and HT tests, we developed a standardized form to record start and end times of key continuous procedural steps for the respective tests from sample preparation to result report (Supporting information).

## Cost analysis

We conducted our costing study based on activity-based, bottom-up costing method as suggested by Sohn et al., 2009 [12]. Cost data was classified into five resource categories: equipment, test kit (consumables), staff, building, and overhead costs (space, overhead staff). We used a range of potential equipment and test kit cost (inclusive of 20% price mark-up to account for procurement and installation). We fixed the equipment price of the NAATs at $100,000 (Range: $50,000 - $100,000). For test kits, we used $13 (Range: $11 - $15) and $9 ($6.6 - $12) per-test estimates as provided by the manufacturers for their pricing structure for the respective NAATs. Capital asset costs were annuitized based on expected useful life years (ELY)– 10 years for laboratory equipment and 30 years for building spaces [12–14]–and a 3% discount rate, plus a 5% annual mark-up to account for the maintenance costs. All costs were captured in 2019 US dollars.

As we did not have access to real-world laboratory operations data to construct per-test costs, we made assumptions for key laboratory operational and cost elements based on simplified approximations from earlier internal FIND studies conducted in LMICs [15–20]. We assumed that the candidate laboratory would operate 250 days annually with 8 hours of work per day. This estimate was varied by 20% for operational days and 25% for hours per day in the sensitivity analyses. Laboratory technician's salary (PE: $5,000/year) and laboratory overhead costs (PE: $65,500/year), inclusive of utilities, furniture and other equipment, and management of admin and staff, were varied by 50% to account for a wide range of LMIC settings. To account for building costs, we assumed that 15 m$^2$ (Range: 8–23 m$^2$) was needed to implement molecular testing capacity with the cost of building space at $1,000 per m$^2$ (Range: $500-$1500). We assumed that candidate laboratories for moderate complexity automated NAATs typically dedicate 50% of their workloads for TB. Complete list of parameters with ranges are presented in Table 1.

We set the equipment price to $100,000, requiring a 5% overhead for maintenance, with an expected useful life of 10 years at a 3% annual discount rate. We assumed that candidate laboratories for moderate complexity automated NAATs typically dedicate 50% of their workloads for TB. We assumed that the candidate laboratory would operate 250 days annually with 8 hours of work per day. This estimate was varied by 20% for operational days and 25% for

**Table 1. Key drivers of per test cost.**

| Key drivers of cost | Primary Estimate | Low | High |
|---|---|---|---|
| Maintenance | 5% | 2% | 10% |
| Equipment Price | $100,000 | %50,000 | $150,000 |
| Discount Rate | 3% | 1% | 10% |
| ELY | 10 | 5 | 15 |
| % TB workload | 50% | 5% | 100% |
| # of TB testing days | 250 | 52 | 300 |
| Approximate Annual Operational days | 250 | 200 | 300 |
| Hours per operating day | 8 | 6 | 10 |
| Laboratory space (m2) for MD | $15 | $7.5 | $22.5 |
| Cost of building space per m2 | $1000 | $500 | $1500 |
| Building annual maintenance cost (per m2) | $100 | $50 | $150 |
| Laboratory Technician Salary (Annual) | $5,000 | $2,500 | $7,500 |
| Utilities | $8,000 | $4,000 | $12,000 |
| Furniture & Equipment | $2,500 | $1,250 | $3,750 |
| Management: Admin & Staff | $55,000 | $27,500 | $82,500 |

hours per day in the sensitivity analyses. Laboratory technician's salary (PE: $5,000/year) and laboratory overhead costs (PE: $65,500/year), inclusive of utilities, furniture and other equipment, and management of admin and staff, were varied by 50% to account for a wide range of LMIC settings. To account for building costs, we assumed that 15 m$^2$ (Range: 8–23 m$^2$) was needed to implement molecular testing capacity with the cost of building space at $1,000 per m$^2$ (Range: $500-$1500).

## Outcome measures

The main outcome of our analysis was per-test cost for a batch of 24 patient samples for each NAAT. All resource costs, except for test kit costs (assessed on a per-sample basis), were converted into cost per minute-use based on estimated total annual laboratory operating time. These per minute-use costs estimates were multiplied by mean activity-based time estimates for procedure in the TAM study. Human resource costs were assessed based on direct hands-on time and per-minute salary estimates.

## Sensitivity analysis

We carried out two-tiered sensitivity analyses. In the first sensitivity analysis, we set laboratory operations and cost parameter ranges to reflect low (smaller, lower-level laboratories) and high-cost (larger, higher-level laboratories) scenarios for testing 24 patient samples per batch-run. In the second analysis, we conducted a series of sensitivity analyses testing wherein we individually varied the equipment price, discount rate, ELY, percentage of TB workload, testing capacity per day and number of testing days while keeping all other parameters constant, for each of the Primary Estimate (PE), low and high scenarios. We varied the discount rate from 1% to 10%, the share of TB workload from 5% to 100%, the testing days from 26 (once every 2 weeks) to 260 (every weekday), the equipment price from $50,000 to $150,000, and daily testing capacity day from 12 to 24 samples per batch for the LT NAAT and 24 to 96 samples per batch for the HT NAAT. All analyses were performed using Microsoft Excel v15.26.

## Ethics statement and details of informed consent

Phase 2 of the protocol "Multicenter study to assess the performance of centralized assay solutions for detection of MTB and resistance to Rifampicin and Isoniazid" was approved by Human Research Ethics Committee (Medical), University of the Witwatersrand on February 20, 2019. As no human subjects were involved for this hypothetical costing exercise, no informed consent was required.

## Results

### Cost per test

Fig 1 provides estimates of the cost per test by cost category for the primary estimate, low-cost and high-cost scenarios. The base-case estimate which used a batch size of 24 samples, 50% TB workload, an equipment price of $100,000 with a 3% discount rate and an expected useful life of 10 years resulted in a per-test cost for LT NAAT of $18.52 and $15.37 for HT NAAT. The range in cost difference between high and low-cost estimates for each category can be found in the Supporting information. The cost of the test kit is a major constituent of the total per-test cost, and equipment costs are highly sensitive and act as key drivers for the total per-test cost under different sensitivity scenarios. Overhead and staff costs are influenced by direct hands-on-time. The longer the hands-on-time, the more likely it will be for staff and overhead to become key drivers of the overall test cost.

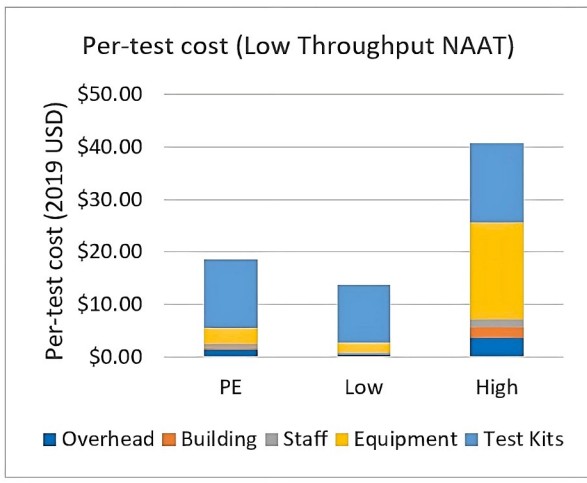
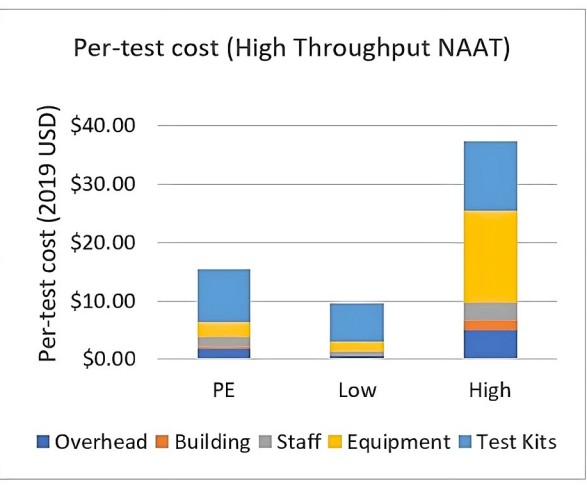

**Fig 1. Cost per test and cost composition for 24-batch Low Throughput NAAT and High Throughput NAAT (Minimum, Maximum, and Primary Estimate).** The height of each colored bar indicates the contribution of a particular component to the overall per test cost. Price per test is highly sensitive to the price of the equipment (NAAT). The cost of the test kit is a major component of the per test cost.

### Sensitivity analysis

We constructed tornado diagram (Fig 2) for both LT NAAT and HT NAAT to ascertain the cost drivers. Once weekly testing as opposed to offering testing services 6 days demonstrated a significant difference in per-test costs for both NAATs, with the latter being cheaper. The equipment price and the durability of the equipment also led to a large variance in per-test costs. TB-specific workload, which was varied from 5% to 100%, was also an important cost driver.

Based on the tests conducted per day and the number of testing days annually, calculated a range for the total number of samples that could be processed annually by each NAAT under various operating scenarios and evaluate how the cost per-test varied with annual testing volume, as shown in Fig 3.

Per-test costs were highly sensitive to equipment utilization up to the annual testing volume of 5,000. Unit costs estimates stabilized around 5,000 to 8,000 annual testing volume. Beyond 8,000 annual tests, unit cost estimates were indifferent, and HT NAAT had clear cost advantage over LT NAAT. At such large scale, the main difference between the LT NAAT and HT NAAT costs was attributable to the differences in the test kit prices, which was generally lower for HT NAAT (approximately $4). The per-test cost of LT NAAT was cheaper than that of HT NAAT if laboratories can procure main testing equipment for the LT NAAT at lower cost (50% in our analysis) than that of HT NAAT, and when HT NAAT equipment is less durable. These factors were most sensitive in lower testing volumes (less than 3,000 per year) than at higher testing volumes (Fig 3).

### Discussion

This study was conducted as a companion piece to the 2020 WHO guidance on TB diagnostics and compares the cost per-test of the LT NAAT and HT NAATs in the context of TB. It is one of the first studies that explores the cost of different diagnostic NAATs for drug-susceptible and drug-resistant TB, making it a valuable contribution to the TB diagnostics literature. Through the time-and motion analysis we were able to get a detailed breakdown of the procedure time and subsequent price per-test for both NAATs. The per-test cost for the LT NAAT was $18.52 (range: $13.79 - $40.70) and $15.37 for the HT NAAT (range: $9.61 - $37.40).

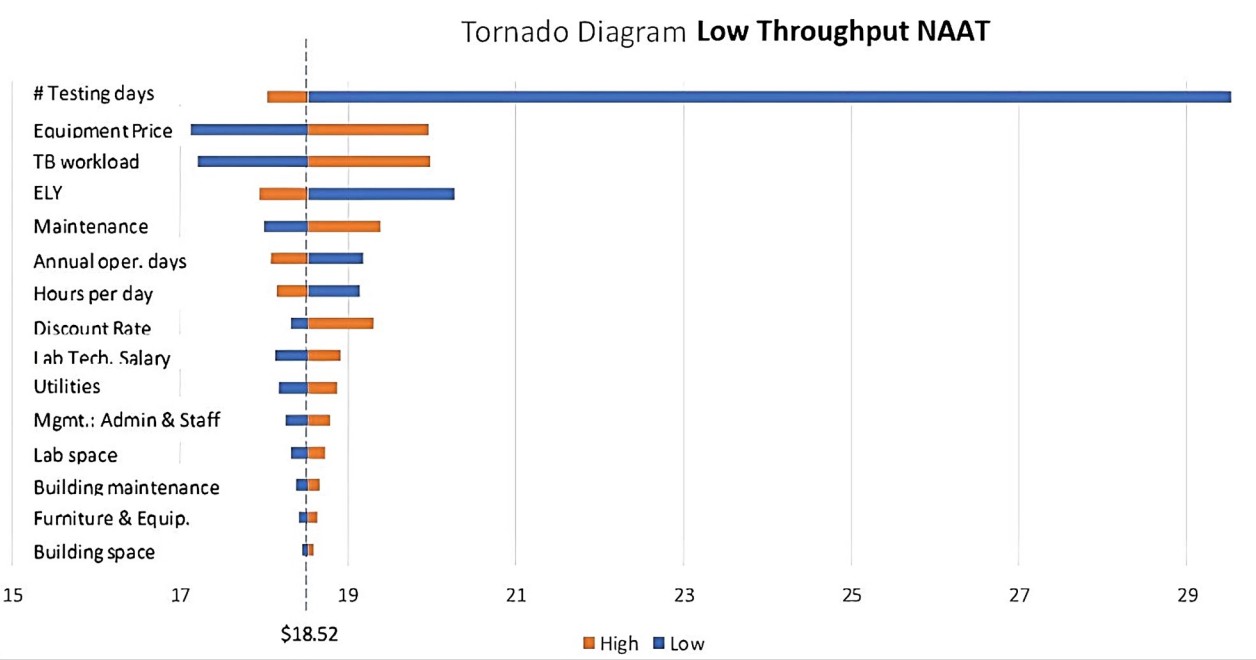

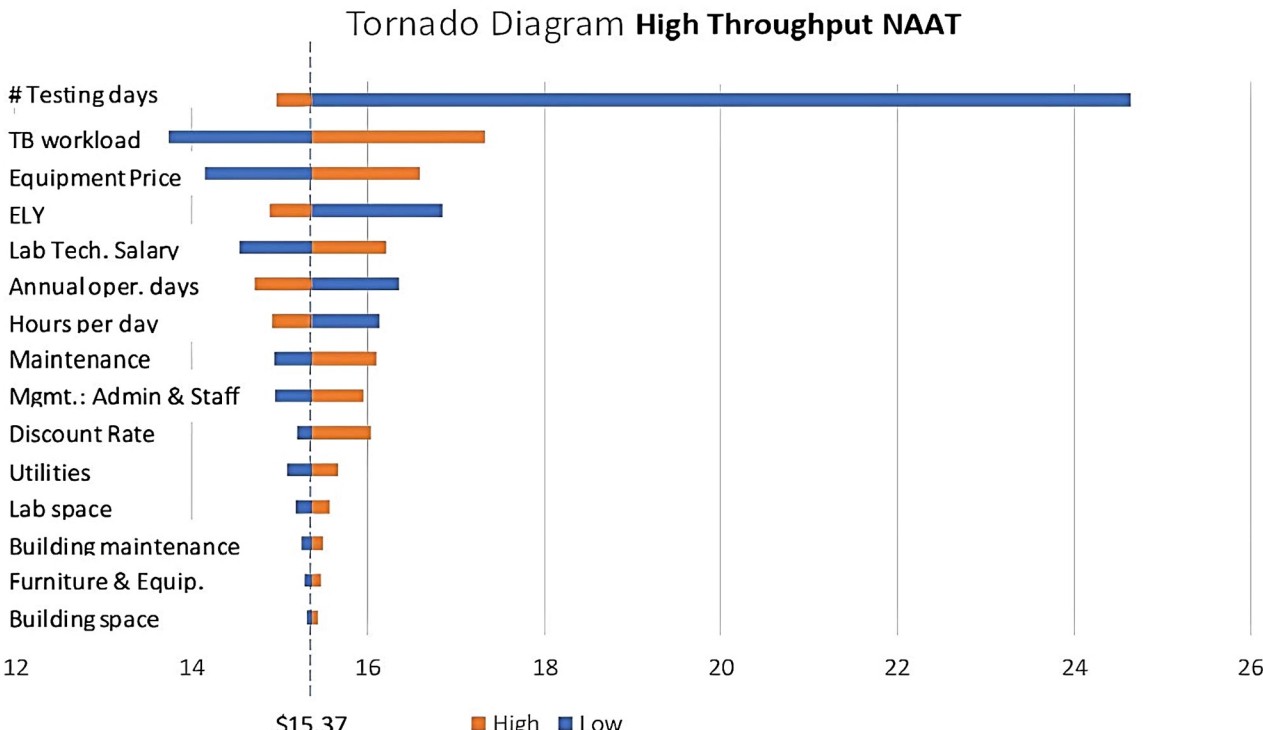

**Fig 2. Tornado plot illustrating key drivers of per test cost for Low Throughput NAAT and High Throughput NAAT.** The reference line in both the plots represent the base case cost when all the parameters were set to their primary estimate values. The width of the bar indicates the change in per test cost with change in the value of the respective parameter. The blue bars indicate change in per test cost when the value of the respective parameter is lesser than the primary estimate. The orange bars indicate change in per test cost when the value of the respective parameter is more than the primary estimate. The drivers of cost are arranged in descending order of their influence on the per test cost.

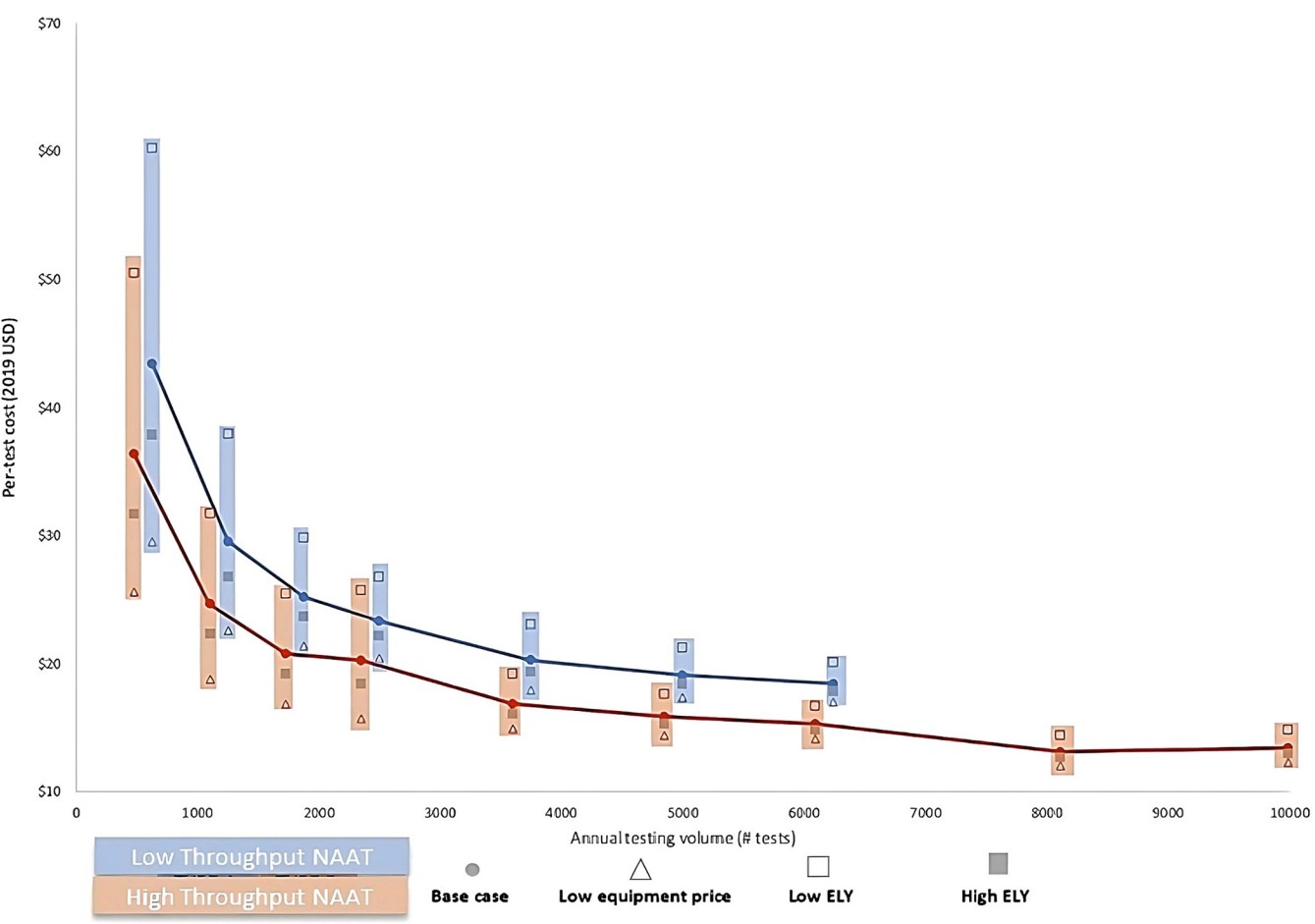

**Fig 3. Per test cost by annual testing volume for Low Throughput NAAT and High Throughput NAAT.** The vertical bars represent variance in per test cost at equal testing volumes for both Low Throughput (LT) NAAT and High Throughput (HT) NAAT. The shapes superimposed on the vertical bars indicate the costs at different scenarios. The bars for HT NAAT have been shifted vertically to the left to make the scenarios for both LT NAAT and HT NAAT legible. LT NAAT stops at 6240 tests because this is the estimated maximum possible service volume per year (24 samples per day, 260 days). At annual testing volumes less than 5000 tests, per-test cost is highly sensitive against equipment utilization for both testing platforms. At annual testing volumes between 5,000 and 8,000 tests, unit-cost starts to stabilize. Beyond 8,000 tests annually (equipment highly utilized), unit-cost estimate is robust against most parameters.

At similar conditions, the main difference for the per-test cost is attributable to differences in the test kit cost. Across the PE, Low-cost, and High-cost scenarios, the difference in the per-test cost of each NAAT is around $3-$4, like the difference in the unit price of the test kit: $13 for LT NAAT (range: $11 - $15) and $9 for HT NAAT (range: $6.6 - $12). If we exclude the cost of the testing kit, the operational per-test cost for the LT NAAT drops to $5.52, lower than HT NAAT at $6.37. Efforts aimed at reducing the test kit cost of LT NAAT will make it a more viable option for testing, especially at low volume scenarios.

The reason LT NAAT has a lower operational cost, despite a higher total per-test cost, is due to the overhead, building and staff costs. These costs were influenced by the direct hands-on-time for testing. Direct hands-on staff time accounts for more than 25% of total run time for the HT NAAT, while only around 11% of the testing time for LT NAAT required direct staff involvement, leading to higher operational costs for HT NAAT. It would be worth exploring further efficiencies that could reduce the staff time for the HT NAAT, which will reduce the per-test cost. Additionally, when considering other testing platforms that require longer run times or more direct hands-on time, these operational expenses will become important drivers of cost.

To make a comparable analysis between LT NAAT and HT NAAT, we assumed 24 tests were performed per batch and no adjustments were made for the losses due to underutilization of the HT NAAT, which has the capacity to complete 96 tests per batch. If under the same conditions, a 96-test batch is run on HT NAAT, the cost per-test would decrease from $15.37 to $12.10. In some high-volume facilities, multiple batches of tests may be run per day, which could further decrease the cost of testing.

In the sensitivity analysis, the biggest drivers of cost were service volume, equipment costs, and equipment durability. At testing volumes of less than 5,000 total tests annually, the per-test cost is highly sensitive to equipment utilization and small changes in service volume led to large changes in the cost per test. The per-test cost becomes more robust as we increase the number of tests annually between 5,000 and 8,000 and becomes nearly constant and robust at volumes higher than 8,000 tests per year. While the current study focused on the cost of these NAATs regarding TB testing, more specifically drug-susceptibility testing for DR-TB. Early diagnosis and optimal treatment of isoniazid-resistant TB may limit emergence of MDR-TB during treatment [21]. Higher testing volumes at centralized labs may also be attained by multiplexing the equipment for use in detecting other diseases such as STIs, SARS-CoV-2, and HPV.

Like most studies, ours also has some limitations. First, the time estimates from the time and motion study were calculated using a low sample size, suggesting higher variability in the estimates. Second, the equipment costs are fixed at $100,000, rather than the estimates provided by manufacturers. This was done intentionally to improve the comparability of the tests and better evaluate how variability in equipment costs can impact price, and the sensitivity analyses capture a wide range of equipment prices. The costs include not just the price of capital assets, but also accounts for installation and maintenance costs. Third, we did not account for costs associated with differences in test performance (accuracy), rate of repeat testing (due to failed or indeterminate test results), and other operational factors such as turn-around-times against the current standard DST. Regarding the test performance, the parent study reported equivalent test accuracies for MC-NAATs compared to existing molecular DST tests (e.g., Xpert MTB/RIF and Line Probe Assays, LPA) [11]. Turn-around-times for MC-NAAT, while theoretically being a same-day test (i.e. full testing time is less than 4 hours in total from sample processing to result), can depend heavily on how frequently can the laboratory run MC-NAAT tests (per week or per month). In very low volume laboratories (i.e. once weekly or bi-weekly testing, corresponding to annual testing volumes less than 1,500 samples), there may not be any benefits in turn-around-times nor in terms of per-test costs relative to the conventional DST.

Placement for MDR-TB diagnostics should be informed by a strategic cost-effective approach that considers logistics outside of the laboratory as well, including but not limited to sample transport related costs, presence of staff with requisite training, access to favorable reagent deals, multiplexing capabilities, and service volume. There is a push to make TB-NAAT testing accessible via single module machines at lower level/decentralized health facilities, to increase point of care testing and reduce loss to follow up at the early diagnosis stage [22]. However, such an intervention may not be cost-effective for all TB programs, especially in a low population setting with small testing volumes.

Centralized testing requires an efficient sample transportation network which would include costs attributed to staff time, fuel, and sample storage, and costs for implementation of such a network in regions where no such infrastructure currently exists. In addition, the need to transport samples to the central facility and wait for results can lead to gaps in the TB care cascade, leading to patients being lost to follow up at initial stages of TB diagnosis and delaying treatment. This results in poor health outcomes such as under detection or delayed detection of both drug-susceptible and drug-resistant TB.

## Conclusion

Placement decisions of new TB diagnostics technologies must consider a range of local factors including disease burden, health systems infrastructure, availability of affordable reagents, multiplexing capabilities, capacity (e.g., sample transport network, result report and linkage to care programs), demand (i.e., expected testing volume), implementation issues (e.g., laboratory infrastructural needs, staff capacity and training requirement), and costs, both fixed and variable comprehensively from the perspective of the health system, to ensure stronger diagnosis rates of both DS and DR-TB.

## Supporting information

**S1 Table. Mean/median duration time per procedure for Low Throughput NAAT and High Throughput NAAT.** For each discrete activity, two separate time estimates were measured: direct hands-on time of laboratory personnel performing the test and full procedural step to account for laboratory instrument use. As we were not able to measure activity times for various sample batch sizes, we estimated times required to process lower (12 or 24 samples per batch) or higher (96 samples per batch) batch sizes based on multiple measurement (at least three) of typical batch size processed during the FIND's external laboratory evaluation study. In estimating times required to process lower or higher batch sizes (vs. observed), we first calculated per-sample processing times for testing steps for which procedural and hands-on time could vary based on the batch size and calculated estimated total time for each step. (DOCX)

**S1 Fig. Range of unit cost difference across resource types (Low to High).** The size of the bar indicates relative contribution of a cost component to the cost difference range. (JPG)

**S1 Data.** (XLSX)

## Acknowledgments

We would like to acknowledge the contributions of Wendy Susan Stevens and Lesley E Scott, University of Witwatersrand, who oversaw the time-and-motion study that assisted us with certain cost estimates.

## Author Contributions

**Conceptualization:** Margaretha De Vos, Hojoon Sohn.

**Data curation:** Akash Malhotra, Anura David, Hojoon Sohn.

**Formal analysis:** Akash Malhotra, Ryan Thompson, Hojoon Sohn.

**Methodology:** Akash Malhotra, Hojoon Sohn.

**Supervision:** Hojoon Sohn.

**Visualization:** Akash Malhotra.

**Writing – original draft:** Akash Malhotra, Ryan Thompson, Hojoon Sohn.

**Writing – review & editing:** Margaretha De Vos, Anura David, Samuel Schumacher, Hojoon Sohn.

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
