## [Decision Letter · Decision Letter 0]

7 Mar 2023

PONE-D-22-29729DETERMINING COST AND PLACEMENT DECISIONS FOR MODERATE COMPLEXITY NAATs FOR TUBERCULOSIS DRUG SUSCEPTIBILITY TESTINGPLOS ONE

Dear Dr. Sohn

Thank you for submitting your manuscript to PLOS ONE. After careful consideration, we feel that it has merit but does not fully meet PLOS ONE’s publication criteria as it currently stands. Therefore, we invite you to submit a revised version of the manuscript that addresses the points raised during the review process.

ACADEMIC EDITOR: 

Major Revision Required 

We look forward to receiving your revised manuscript.

Kind regards,

Divakar Sharma

Academic Editor

PLOS ONE

“This work was funded through multi-donor core funding to Foundation for Innovative New Diagnostics (FIND). We would like to also acknowledge the contributions of Wendy Susan Stevens and Lesley E Scott, University of Witwatersrand, who oversaw the time-and-motion study that assisted us with certain cost estimates. This work was also supported by the New Faculty Startup Fund from Seoul National University (HS)”

5. Please amend the manuscript submission data (via Edit Submission) to include author Lesley E Scott.

Reviewers' comments:

Reviewer's Responses to Questions

**Comments to the Author**

1. Is the manuscript technically sound, and do the data support the conclusions?

Reviewer #1: Yes

Reviewer #2: No

Reviewer #3: Yes

2. Has the statistical analysis been performed appropriately and rigorously? 

Reviewer #1: Yes

Reviewer #2: No

Reviewer #3: Yes

3. Have the authors made all data underlying the findings in their manuscript fully available?

Reviewer #1: Yes

Reviewer #2: Yes

Reviewer #3: Yes

4. Is the manuscript presented in an intelligible fashion and written in standard English?

Reviewer #1: Yes

Reviewer #2: Yes

Reviewer #3: Yes

5. Review Comments to the Author

Reviewer #1: In this paper Malhotra et al conducted a cost analysis of two groups of moderate complexity NAATs with different testing throughput: Lower Throughput (LT, < 24 tests per run) and Higher Throughput (HT, upto 90+ tests per run). This cost analysis study is of great importance to guide and assist the options and placement of assays and systems in resource limited settings. Authors used per-test cost as the main indicator to assess the drivers of cost by resource types and optimized levels of annual testing volumes for the respective NAATs. The authors reached base-case per test costs of $18.52 (range: $13.79 - $40.70) for LT test and $15.37 (range: $9.61 - $37.40) for HT test. Authors assumed equivalent performance and infrastructural needs, for group-comparative purposes.

Its a well grounded, well preformed and consistent work. The conclusions are supported by the results. This work provides important information especially for LMIC TB countries and TB programs in the decision making regarding new molecular technology implementation. The paper brings relevant information to science, especially implementation science applied to TB laboratories in LMIC and for the management of M/XDRTB.

There are few aspects that need to be improved:

- In the introduction the advantages and limitations of molecular biology based testing incorporation in TB programs needs to be addressed and reviewed, especially accuracy/performance limitations, connection with clinical outcomes and program aims and cost-effectiveness. The criteria to place these systems in primary care vs centralized labs and TB networks, related with the market target of LT avd HT systems ins not explored nor presented. This subject is relevant and is very well addressed in the WHO guidelines from where this study is a helpfull spin-off.

- In the materials and methods the authors should clearly define which molecular systems endorsed by WHO were included in the Low Throughput (LT) group and which were included in the High Throughput (HT). For a non expert its difficult to differentiate the inclusion criteria for each of the groups, the available and endorsed systems and another concept also used in the paper- MC-NAATS. The paper as it is, is difficult to connect the results with, at least, the canonical examples of WHO endorsed LT and HT systems, as defined by the authors. Please define better the groups inclusion criteria and give examples of systems included.

- The references in the text use different formats. Please check and correct.

- The quality of the graphs is poor and should follow PlosOne authors guidelines for figures.

Reviewer #2: In this article, the authors studied the low- and high throughput MC-NAAT for TB DST. This is an assumption-based study and the authors report that the cost effectiveness depends on the testing volume. I think this is a pre-mature study, and the real-world trend could be very different. Hence, it is advised to test their assumptions at least at a minimal TB testing centers and compare the cost, turnaround time, reliability of the results with available standard DST testing techniques.

Reviewer #3: Overall comments

This research paper deals with an important subject of determining the conditions under which per-test costs can be optimized for each NAAT and evaluating the drivers of the cost. Overall this is a good work but the authors have to address few queries and provide few clarifications.

Specific Comments

• The authors have used two hypothetical laboratories in a resource-limited setting, is it the standard methodology followed in studies of this kind? Authors can give references for this to strengthen the methodology.

• I suggest the authors to provide some more details on automated NAATs for a proper understanding to the readers

• It has been mentioned that the current study assessed the costs of two moderate complexity automated NAATs, what were these two NAATs?

• Capital asset useful life years - 10 years for laboratory equipment and 30 years for building spaces- provide reference for this statement

• Why have the authors used different discount rates 3% 1% 10% in the study as the discount rates will not change.

• All costs were captured in 2019 US dollars; if possible, authors can convert it to the current value for the years 2022 or 2023.

• It would be good if the authors provide methodology with sub-titles such as model structure, parameters used, data collection and source, and analysis

• Primary estimate (PE) to be estimated in the figures

• In figure two, the lower value for testing days seems to be very high, please check

6. PLOS authors have the option to publish the peer review history of their article (what does this mean?). If published, this will include your full peer review and any attached files.

Reviewer #1: No

Reviewer #2: No

Reviewer #3: No

---

## [Author Response · Author response to Decision Letter 0]

25 Jul 2023

Responses to the editor and reviewers

Answer: Thank you for directing us to the relevant links. We have made the required formatting changes across the manuscript. Please also note that the earlier submitted file “S1 Appendix” has now been uploaded separately as “S1 Table” and “S1 figure” with the requested format. 

Answer: Thank you for highlighting this. In the Ethics statement and details of informed consent section, we mentioned that “no human subjects were involved for this hypothetical costing exercise”. This can be found in line 143-144 of the document. The observation only captured the workflow for each of the constituent tests during a testing cycle and did not observe any human subjects. 

“This work was funded through multi-donor core funding to Foundation for Innovative New Diagnostics (FIND). We would like to also acknowledge the contributions of Wendy Susan Stevens and Lesley E Scott, University of Witwatersrand, who oversaw the time-and-motion study that assisted us with certain cost estimates. This work was also supported by the New Faculty Startup Fund from Seoul National University (HS)”

Answer: Thank you for the appropriate guidance. We have now removed all funding information from the Acknowledgements or other sections of the manuscript. We have reflected this information in our new cover letter. We thank you for editing the online submission on our behalf. 

Answer: We have now uploaded, as supplementary information, the minimal data set underlying the results described in our manuscript. We have reflected this detail in our revised cover letter and thank you for updating the Data Availability Statement. 

5. Please amend the manuscript submission data (via Edit Submission) to include author Lesley E Scott.

Answer: Thank you for pointing this out. Lesley E. Scott had requested to be removed as co-author and we thanked them in the Acknowledgements Section. We had erroneously left their name in the author list and have now removed the same. 

Reviewers' comments:

Reviewer's Responses to Questions

Comments to the Author

1. Is the manuscript technically sound, and do the data support the conclusions?

Reviewer #1: Yes

Reviewer #2: No

Reviewer #3: Yes

2. Has the statistical analysis been performed appropriately and rigorously?

Reviewer #1: Yes

Reviewer #2: No

Reviewer #3: Yes

3. Have the authors made all data underlying the findings in their manuscript fully available?

Reviewer #1: Yes

Reviewer #2: Yes

Reviewer #3: Yes

4. Is the manuscript presented in an intelligible fashion and written in standard English?

Reviewer #1: Yes

Reviewer #2: Yes

Reviewer #3: Yes

5. Review Comments to the Author

Reviewer #1: In this paper Malhotra et al conducted a cost analysis of two groups of moderate complexity NAATs with different testing throughput: Lower Throughput (LT, < 24 tests per run) and Higher Throughput (HT, upto 90+ tests per run). This cost analysis study is of great importance to guide and assist the options and placement of assays and systems in resource limited settings. Authors used per-test cost as the main indicator to assess the drivers of cost by resource types and optimized levels of annual testing volumes for the respective NAATs. The authors reached base-case per test costs of $18.52 (range: $13.79 - $40.70) for LT test and $15.37 (range: $9.61 - $37.40) for HT test. Authors assumed equivalent performance and infrastructural needs, for group-comparative purposes.

Its a well grounded, well preformed and consistent work. The conclusions are supported by the results. This work provides important information especially for LMIC TB countries and TB programs in the decision making regarding new molecular technology implementation. The paper brings relevant information to science, especially implementation science applied to TB laboratories in LMIC and for the management of M/XDRTB.

There are few aspects that need to be improved:

Answer: We thank you for your encouraging comments and overall feedback. Please see our responses to your specific queries below.

- In the introduction the advantages and limitations of molecular biology based testing incorporation in TB programs needs to be addressed and reviewed, especially accuracy/performance limitations, connection with clinical outcomes and program aims and cost-effectiveness. The criteria to place these systems in primary care vs centralized labs and TB networks, related with the market target of LT avd HT systems ins not explored nor presented. This subject is relevant and is very well addressed in the WHO guidelines from where this study is a helpfull spin-off.

Answer: We appreciate your thoughtful comment. We made necessary revisions to reflect your comments and provided advantages and drawbacks of molecular testing for the centralized assays in question. We have also referenced the WHO guidelines and relevant literature for the same. These revisions are made in lines 56-62: “These automated NAATs offer higher throughput capacity (single-run capacity: 24 to 96 samples) with comparable performance and lower infrastructural needs compared to WHO endorsed DSTs, thus being advantageous to both patients and programs (9,10). These NAATs are faster and less complex than phenotypic culture based drug susceptibility testing (DST) and line-probe assays (LPA), being largely automated after the sample preparation step(9). Placement of these NAATs strategically in relation to the point-of-care can help improve DST coverage and DR-TB patient management(11). However, successful implementation of these NAATs is dependent on a robust sample transport network and a high population density(9). This may potentially lead to exclusion of low population communities with poor infrastructure.”

- In the materials and methods the authors should clearly define which molecular systems endorsed by WHO were included in the Low Throughput (LT) group and which were included in the High Throughput (HT). For a non expert its difficult to differentiate the inclusion criteria for each of the groups, the available and endorsed systems and another concept also used in the paper- MC-NAATS. The paper as it is, is difficult to connect the results with, at least, the canonical examples of WHO endorsed LT and HT systems, as defined by the authors. Please define better the groups inclusion criteria and give examples of systems included.

Answer: Thank you for your comment. In the Intervention sub-section under Methods (lines 75-78) we have defined what we mean by low throughput and high throughput sections. Since this is a hypothetical costing exercise, and is manufacturer agnostic, we did not want to name the specific device(s). Our intention is to only comment on the drivers of cost and placement strategies of these MC NAATs. Revisions are made as the following:

“We defined Low Throughput (LT) system as those test platforms with a capacity to test up to 24 samples per batch (i.e., singe test run). High Throughput (HT) test was defined as a system with a capacity of test up to 96 samples per batch, and which required reflex testing for confirmatory diagnosis.” 

- The references in the text use different formats. Please check and correct.

Answer: Thank you for catching this. We noticed a superscript error for one of the references and have corrected the same. 

- The quality of the graphs is poor and should follow PlosOne authors guidelines for figures.

Answer: Thank you for directing us to the PlosOne guidance for the figures. We have upgraded the quality of our figures (minimum pixel and appropriate size requirements) and have saved them using the correct extension (TIF). As recommended by the editor, these images were uploaded to the PACE digital diagnostic tool for images and now meet the PlosOne requirements. 

Reviewer #2: In this article, the authors studied the low- and high throughput MC-NAAT for TB DST. This is an assumption-based study and the authors report that the cost effectiveness depends on the testing volume. I think this is a pre-mature study, and the real-world trend could be very different. Hence, it is advised to test their assumptions at least at a minimal TB testing centers and compare the cost, turnaround time, reliability of the results with available standard DST testing techniques.

Answer: We thank you for your thoughtful comment. Despite being a hypothetical study, our analyses are based on time estimates and laboratory resources used for the entire testing workflow directly observed for various testing volumes. from our time and motion sub-study. We have performed a range of sensitivity analyses to reflect operations at smaller (or larger) laboratories that has low (minimal testing) and high testing volumes. These scenarios reflect a range of laboratory operations from minimal to high volume testing (see our sensitivity analysis section). As our study was carried as part of an initial evaluation study (i.e., proof of concept study) by FIND, we did not have a comparative data on turn-around-time. Furthermore, our aim was to estimate and compare per-test cost of low and high throughput MC-NAATs, not compare these costs to conventional DST. Therefore, we did not have empiric cost estimates for conventional DST (e.g., LPA or culture). However, diagnostic accuracy of these MC-NAATs were assessed as part of the main study and these results are reported in (REF – FIND MC-NAAT proof of concept study results in JCM). To address the reviewer’s concerns related to this matter, we have added texts in the discussion section as the following: 

Limitations in not being able to compare test performance, turn-around-time, and costs of MC-NAAT to conventional DSTs:

Lines 248-257: “Third, we did not account for costs associated with differences in test performance (accuracy), rate of repeat testing (due to failed or indeterminate test results), and other operational factors such as turn-around-times against the current standard DST. Regarding the test performance, the parent study reported equivalent test accuracies for MC-NAATs compared to existing molecular DST tests (e.g., Xpert MTB/RIF and Line Probe Assays, LPA). Turn-around-times for MC-NAAT, while theoretically being a same-day test (i.e. full testing time is less than 4 hours in total from sample processing to result), can depend heavily on how frequently can the laboratory run MC-NAAT tests (per week or per month). In very low volume laboratories (i.e. once weekly or bi-weekly testing, corresponding to annual testing volumes less than 1,500 samples), there may not be any benefits in turn-around-times nor in terms of per-test costs relative to the conventional DST.”

Costs associated with other operations (e.g., sample transport network)

Lines 258-264: “Placement for MDR-TB diagnostics should be informed by a strategic cost-effective approach that considers logistics outside of the laboratory as well, including but not limited to sample transport related costs, presence of staff with requisite training, access to favorable reagent deals, multiplexing capabilities, and service volume. There is a push to make TB-NAAT testing accessible via single module machines at lower level/decentralized health facilities, to increase point of care testing and reduce loss to follow up at the early diagnosis stage(19). However, such an intervention may not be cost-effective for all TB programs, especially in a low population setting with small testing volumes. 

Centralized testing requires an efficient sample transportation network which would include costs attributed to staff time, fuel, and sample storage, and costs for implementation of such a network in regions where no such infrastructure currently exists. In addition, the need to transport samples to the central facility and wait for results can lead to gaps in the TB care cascade, leading to patients being lost to follow up at initial stages of TB diagnosis and delaying treatment. This results in poor health outcomes such as under detection or delayed detection of both drug-susceptible and drug-resistant TB.” 

Reviewer #3: Overall comments

This research paper deals with an important subject of determining the conditions under which per-test costs can be optimized for each NAAT and evaluating the drivers of the cost. Overall this is a good work but the authors have to address few queries and provide few clarifications.

Specific Comments

• The authors have used two hypothetical laboratories in a resource-limited setting, is it the standard methodology followed in studies of this kind? Authors can give references for this to strengthen the methodology.

Answer: Thank you for your comment. While ours is a hypothetical study, we structured our cost analysis based on the costing guideline for TB (and other) diagnostic tests that was developed by one of our co-authors (H Sohn). We have revised the text in the beginning of the ‘Cost analysis’ section in the methods to reflect this: 

Line 83-84: “We conducted our costing study based on activity-based, bottom-up costing method as suggested by by Sohn et al., 2009”

As such, we follow-up activity-based, bottom-up costing method where we use time and motion data to define duration of key resource use (e.g., laboratory equipment, human resource, and laboratory space) and actual use of key laboratory (test kits, reagents, etc.) and general consumables (laboratory gloves, pipet tips, etc.). As the parent study was conducted in a single laboratory in South Africa, we were not able to ascertain overhead costs of different ‘types’ of laboratories in a range of low-and-middle-income country contexts. As such, we generated a range of assumptions on costs of overall laboratory overheads, human resource costs, and other operational costs of different types of laboratories where MC-NAATs could be placed in the LMIC context. We test these range of assumptions to demonstrate the robustness of our primary per-test costs and suggest different scenarios in which lower or higher per-test cost estimates may be relevant. 

• I suggest the authors to provide some more details on automated NAATs for a proper understanding to the readers

Answer: Thank you for your suggestion. We have expanded on our description of automated NAATs in following lines: 

Lines 56-59: “thus being advantageous to both patients and programs (9,10). These NAATs are faster and less complex than phenotypic culture based drug susceptibility testing (DST) and line-probe assays (LPA), being largely automated after the sample preparation step(9)”

And… 

Lines 75-78: “We defined Low Throughput (LT) system as those test platforms with a capacity to test up to 24 samples per batch (i.e., singe test run). High Throughput (HT) test was defined as a system with a capacity of test up to 96 samples per batch, and which required reflex testing for confirmatory diagnosis.” 

• It has been mentioned that the current study assessed the costs of two moderate complexity automated NAATs, what were these two NAATs?

Answer: Since this is a hypothetical costing exercise, and is manufacturer agnostic, we did not want to name the specific device(s). Our intention is to only comment on the drivers of cost and placement strategies of MC NAATs which were featured in the 2021 WHO Diagnostics Guidelines. Instead, we have provided specifics of these two test platforms based on test throughput (low and high): 

Lines 75-78: “We defined Low Throughput (LT) system as those test platforms with a capacity to test up to 24 samples per batch (i.e., singe test run). High Throughput (HT) test was defined as a system with a capacity of test up to 96 samples per batch, and which required reflex testing for confirmatory diagnosis.”

• Capital asset useful life years - 10 years for laboratory equipment and 30 years for building spaces- provide reference for this statement

Answer: Thank you for catching this. We have provided a reference for these values. 

• Why have the authors used different discount rates 3% 1% 10% in the study as the discount rates will not change.

Answer: Thank you for your comment. In the real-world context, discount rates may vary. However, for analytical purposes, economic evaluations, including cost analyses, use fixed discount rate for primary estimates. For our analysis, we have used a discount rate of 3% for our primary estimate then varied the discount rates between 1 and 10% to test for uncertainties in our per-test cost estimate, which did not change considerably. 

• All costs were captured in 2019 US dollars; if possible, authors can convert it to the current value for the years 2022 or 2023.

Answer: Thank you for your suggestion. With all due respect, while it is possible to cover our estimates to 2022 or 2023 US dollars, generally it would be best to provide cost estimates that align with the dates/years of the primary study. Hence, we would like to keep our cost estimates as 2019 US dollar figure. 

• It would be good if the authors provide methodology with sub-titles such as model structure, parameters used, data collection and source, and analysis

Answer: We appreciate your suggestion and have added sub-titles throughout. 

• Primary estimate (PE) to be estimated in the figures

Answer: Thank you for your comment. In our figure captions for all 3 of our figures, we have specified which values/symbols refer to the primary estimate or base case. Please let us know if we are missing anything and we will be able to rectify the same. 

• In figure two, the lower value for testing days seems to be very high, please check

Answer: Thank you for requesting us to check this. The “Low” value for testing days seems high because of the relative difference in the number of testing days. In the “Low” scenario, we assumed testing for one day a week whereas in the primary estimate we assumed testing for 5 days a week, as specified in table 1. 

6. PLOS authors have the option to publish the peer review history of their article (what does this mean?). If published, this will include your full peer review and any attached files.

Do you want your identity to be public for this peer review? For information about this choice, including consent withdrawal, please see our Privacy Policy.

Reviewer #1: No

Reviewer #2: No

Reviewer #3: No

While revising your submission, please upload your figure files to the Preflight Analysis and Conversion Engine (PACE) digital diagnostic tool, https://pacev2.apexcovantage.com/. PACE helps ensure that figures meet PLOS requirements. To use PACE, you must first register as a user. Registration is free. Then, login and navigate to the UPLOAD tab, where you will find detailed instructions on how to use the tool. If you encounter any issues or have any questions when using PACE, please email PLOS at figures@plos.org. Please note that Supporting Information files do not need this step

Answer: Thank you for this tip. The images were uploaded and then downloaded from the PACE tool after correction.

---

## [Decision Letter · Decision Letter 1]

10 Aug 2023

DETERMINING COST AND PLACEMENT DECISIONS FOR MODERATE COMPLEXITY NAATs FOR TUBERCULOSIS DRUG SUSCEPTIBILITY TESTING

PONE-D-22-29729R1

Dear Dr. Sohn

We’re pleased to inform you that your manuscript has been judged scientifically suitable for publication and will be formally accepted for publication once it meets all outstanding technical requirements.

Kind regards,

Divakar Sharma Ph.D.

Academic Editor

PLOS ONE

Additional Editor Comments (optional):

Accept

Reviewers' comments:

Reviewer's Responses to Questions

**Comments to the Author**

1. If the authors have adequately addressed your comments raised in a previous round of review and you feel that this manuscript is now acceptable for publication, you may indicate that here to bypass the “Comments to the Author” section, enter your conflict of interest statement in the “Confidential to Editor” section, and submit your "Accept" recommendation.

Reviewer #1: All comments have been addressed

Reviewer #2: (No Response)

Reviewer #3: All comments have been addressed

2. Is the manuscript technically sound, and do the data support the conclusions?

Reviewer #1: Yes

Reviewer #2: (No Response)

Reviewer #3: Yes

3. Has the statistical analysis been performed appropriately and rigorously? 

Reviewer #1: Yes

Reviewer #2: (No Response)

Reviewer #3: Yes

4. Have the authors made all data underlying the findings in their manuscript fully available?

Reviewer #1: Yes

Reviewer #2: (No Response)

Reviewer #3: Yes

5. Is the manuscript presented in an intelligible fashion and written in standard English?

Reviewer #1: Yes

Reviewer #2: (No Response)

Reviewer #3: Yes

6. Review Comments to the Author

Reviewer #1: The authors made all the necessary improuvements and implemented reviewers recomendations as per requested and recomended.

The paper is now ready to be publised.

Reviewer #2: (No Response)

Reviewer #3: The authors have addressed all the comments raised in a satisfactory way. Hence the manuscript can be accepted for publication

7. PLOS authors have the option to publish the peer review history of their article (what does this mean?). If published, this will include your full peer review and any attached files.

Reviewer #1: No

Reviewer #2: No

Reviewer #3: No

---

## [Editor Report · Acceptance letter]

15 Aug 2023

PONE-D-22-29729R1 

Determining cost and placement decisions for moderate complexity NAATs for tuberculosis drug susceptibility testing 

Dear Dr. Sohn:

I'm pleased to inform you that your manuscript has been deemed suitable for publication in PLOS ONE. Congratulations! Your manuscript is now with our production department. 

Kind regards, 

on behalf of

Dr. Divakar Sharma 

Academic Editor

PLOS ONE